# ULNeF: Untangled Layered Neural Fields for Mix-and-Match Virtual Try-On

**Igor Santesteban**
Universidad Rey Juan Carlos
Madrid, Spain
igor.santesteban@urjc.es

**Miguel A. Otaduy**
Universidad Rey Juan Carlos
Madrid, Spain
miguel.otaduy@urjc.es

**Nils Thuerey**
Technical University of Munich
Germany
nils.thuerey@tum.de

**Dan Casas**
Universidad Rey Juan Carlos
Madrid, Spain
dan.casas@urjc.es

## Abstract

Recent advances in neural models have shown great results for virtual try-on (VTO) problems, where a 3D representation of a garment is deformed to fit a target body shape. However, current solutions are limited to a single garment layer, and cannot address the combinatorial complexity of mixing different garments. Motivated by this limitation, we investigate the use of neural fields for mix-and-match VTO, and identify and solve a fundamental challenge that existing neural-field methods cannot address: the interaction between layered neural fields. To this end, we propose a neural model that untangles layered neural fields to represent collision-free garment surfaces. The key ingredient is a neural untangling projection operator that works directly on the layered neural fields, not on explicit surface representations. Algorithms to resolve object-object interaction are inherently limited by the use of explicit geometric representations, and we show how methods that work directly on neural implicit representations could bring a change of paradigm and open the door to radically different approaches.

## 1 Introduction

In virtual try-on (VTO) applications, a computer model of a 3D garment is displayed together with an avatar of a person, to communicate how the garment deforms based on the shape and pose of the person. Neural processing methods have shown great success to solve the problem of VTO [38, 57, 50, 23, 5, 84], by leveraging low-dimensionality parameterizations of body shape and pose [36, 29]. These methods train a 3D deformation model of one garment (or a predefined outfit), and provide an accurate approximation of physics simulation at runtime, while requiring just a small fraction of the computational cost of physical simulations.

However, state-of-the-art VTO is limited to wearing a *single* garment or a predefined outfit, but in real life we combine many clothes to create different outfits. Unfortunately, existing garment-specific or outfit-specific neural processing solutions cannot address the combinatorial complexity of mix-and-match VTO. In fact, the problem of mix-and-match VTO poses novel challenges to machine learning algorithms, as it drives the attention toward object *interaction* problems where each object (*e.g.*, each garment) is geometrically complex, and the space of object-object interactions cannot be exhaustively trained.

36th Conference on Neural Information Processing Systems (NeurIPS 2022).

Object-object interaction has been solved traditionally using explicit geometric representations [43, 2]. Recent advances in neural models of implicit representations have enabled radically new solutions to many problems, with the ability to efficiently encode parametric models [55, 15, 1, 10, 48, 27, 72]. However, for object-object interaction, the applicability of neural implicit models is still limited to solving proximity queries against explicit representations, and interaction is actually solved on these explicit representations [83].

Motivated by the challenges of mix-and-match VTO, we introduce a novel approach to resolve multi-object interaction problems, which works directly on implicit representations of the objects. We represent multiple possibly colliding objects (*e.g.*, multiple garments) using a layered variant of neural fields [75], and we design an algorithm that untangles these layered neural fields to represent collision-free objects. In Section 3 we describe how layered neural fields can be parameterized by deformation codes to represent multiple deformable surfaces, and we introduce a neural untangling projection operator that works directly on the layered neural fields, not on surface representations. The result of compositing the neural projection with the layered neural fields is *untangled layered neural fields* (ULNeFs).

In Section 4 we show how ULNeFs can be used to efficiently solve mix-and-match VTO for multiple garment layers. As a preprocess, each garment layer is represented using a parametric neural field that is trained for this garment in isolation. At runtime, given a code that represents body shape, we optimize for the deformation of the garment layers that is free of collisions, using ULNeFs as the fundamental tool to resolve garment collisions. We demonstrate challenging mix-and-match VTO examples with multiple layers of clothing resolved interactively.

In summary, our contribution is threefold:

1. A neural field formulation, based on covariant fields, capable of encoding open surfaces with holes, as well as inside-outside information (Section 3.1).
2. A neural projection operator that directly projects entangled surfaces, encoded as neural fields, to untangled configurations, coined as ULNeF (Section 3.2).
3. A downstream task using ULNeFs to enable interactive and accurate mix-and-match VTO (Section 4).

ULNeFs could see applicability in other object-object interaction problems beyond mix-and-match VTO. Algorithms to resolve object-object interaction are inherently limited by the use of explicit geometric representations. We show how methods that work directly on implicit representations could bring a change of paradigm and open the door to radically different approaches. Moreover, novel neural-field representations are hard to leverage in situations that involve object interaction and/or collisions, but collision-handling methods rooted on implicit representations could further extend their applicability.

## 2 Related Work

**Neural Fields.** Over the last few years, neural fields [75] have emerged as an alternative to the well-established polygonal meshes to represent shapes. Neural fields build on the –also well-established– idea that shapes can be represented as a level set of implicit functions, but propose that such function can be learned with a neural network [11, 39, 40]. These neural representations are compact, continuous, and are easily differentiable, which opens the door to a large variety of applications in many fields including Computer Vision [54, 85], Computer Graphics [62, 42, 18, 78], and Robotics [61, 65, 47].

Many works leverage the neural fields capabilities to encode human-related features for a variety of tasks. This has enabled, for example, impressive advances in *reconstructing* clothed humans directly from RGB [54, 25, 80], RGB-D [35, 64, 82, 17], and point cloud [12] input. Alternatively, some methods use neural fields to ease the fitting of a parametric human model [36, 29] to sparse inputs [71, 6, 7], which yields to detailed 3D reconstructions that can be articulated by the underlying skeleton.

Closer to ours are the works that use neural fields for *modeling* 3D deformable bodies [15, 45, 1, 41, 30, 31] and clothed humans [10, 48, 27, 55, 72]. This is in contrast to previous works that tackle these 3D modeling tasks with explicit mesh-based models for humans [9, 56, 37, 52], which

typically requires accurate surface registrations, and limits the surface details by the mesh topology. A common strategy is to learn dynamic neural fields in a canonical space, reproducing pose-dependent deformations observed in detailed scans [55, 10] or partial depth maps [48, 72, 17]. The learned field is then articulated using forward skinning techniques. Despite the realism of the output deformations, learned fields encode a *single* surface for clothing and body. In contrast, our formulation defines how to mix and untangle different fields to allow the editing of clothing styles.

Neural fields have also been used to encode both appearance and volumetric information of a scene, a representation known as Neural Radiance Fields (NeRF) [42]. Follow up works showed that NeRFs can be used also to encode articulated objects [76, 46, 77] and dynamic scenes [53, 19, 49]. Specific for humans, A-NeRF [63] transforms NeRF features using an skeleton, and demonstrates that novel motions and viewpoints can be synthesized. NeuralBody [51] appends learnable features to the vertices of a surface body model, enabling free-viewpoint rendering of animatable humans. Similarly, Kwon *et al.* [32] enrich a parametric surface human aggregating spatio-temporal density and color information using transformers. Orthogonal to these works, we do not encode appearance or volume density, but propose a novel formulation to allow the untangled combination of garments encoded using layered fields, which enriches existing representations for humans.

**3D Virtual Try-On.** This area of research aims at estimating how a digital representation of a 3D garment deforms as a function of the underlying body shape. Physics-based cloth simulation [3, 43] has been the natural choice to model this problem, however, given the high computational cost of each simulation time step, recent methods have explore learning-based alternatives [73, 57, 50, 5, 23]. These works aim at learning a function that directly outputs a deformed mesh of a garment given an input feature that describes the target body shape. To this end, the common solution is to use per-vertex supervised strategies that leverage large datasets of simulated [22, 57, 50, 4] or reconstructed 3D garments [37, 52]. Notably, recent methods have proposed self-supervised strategies based on physics-inspired losses [5, 58].

All 3D VTO methods described above use an explicit model to encode the garment mesh. Closest to ours is SMPLicit [13], a recent work that encodes parametric garments as unsigned distance fields learned from data. This allows for controllable garment style, but since an *unsigned* field is used it complicates the extraction of isosurface, and does not allow for inside/outside queries to model collisions.

Garment-body collision handling is particularly challenging in learning-based VTO. Since at inference time most methods directly predict garment vertex positions (*i.e.,* there is not explicit collision check), residual regression errors can potentially lead to garment vertices that penetrate the body. To alleviate this, most methods [5, 50, 57] include a term in their loss function to penalize collisions, but generalization of this term beyond the training dataset is difficult. In fact, an expensive postprocessing step to fix collisions is common in most state-of-the-art learning-based methods, but some speed-up methods exist [66]. To circumvent the need for any fix, Santesteban *et al.* [59] propose to learn a generative subspace that encode collision-free garment configurations, but cannot handle more than one layer. In contrast, our ULNeF approach proposes a completely new formulation based on neural fields that can deal with multiple garments.

**Image-Based Virtual Try-On.** Virtual try-on has also been approached using image-based methods. The aim is to generate compelling 2D images of dressed people, without dealing with any 3D model. Early works based pose-dependent interpolation between different images [26] have been outperform with modern techniques based on convolutional neural networks (CNNs) [24, 69]. Subsequent works further improve the quality of the synthesized images [33, 81, 79, 20], solve artifacts by reducing the reliance on 2D segmentation [28, 21], support mix-and-match VTO [44, 34, 14], and synthesize images for arbitrary poses [16, 70].

## 3 Untangled Layered Neural Fields

The core of our work is a method that takes as input $N$ neural fields, which implicitly represent $N$ possibly colliding surfaces, and outputs $N$ projected fields –the ULNeFs– which encode collision-free implicit surfaces and minimize the displacement with respect to the input. By using implicit representations, the surfaces are defined as zero-sets of scalar functions. Then, the untangling

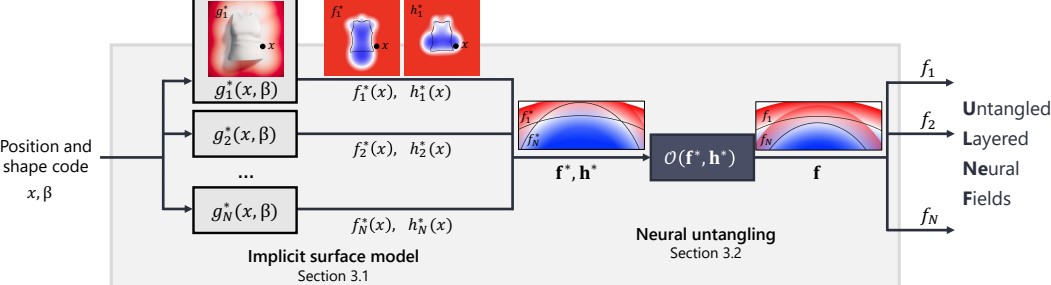

Figure 1: Overview of ULNeF. We propose a formulation to encode parametric open surfaces using a novel neural representation $g_n^*(x, \beta)$ that encapsulates two fields: a *signed distance field* $f(x)$ that represents the garment surface and provides a notion of inside-outside; and a *covariant field* $h(x)$ that models the volume near the openings that other garments can pass through without producing tangled configurations. Having a set of surfaces with potentially entangled configurations, and leveraging their corresponding fields $f_i^*(x)$ and $h_i * (x)$, we propose a neural projection operator that directly deforms the input fields $f_i^*(x)$ such that their zero-sets do not collide.

operation reduces to modifying the scalar functions, which in practice shifts the zero-sets. Figure 1 depicts a summary of our main building block.

In this section, we present the untangling operation as an optimization formulated on scalar field values, and we show how this optimization can be efficiently learned with a neural model. Beforehand, we first discuss specifics of the implicit representation of garments, and how we further parameterize the garments as a function of additional settings, in our case body shape.

## 3.1 Implicit Surface Model

Surfaces can be represented implicitly as the zero-set of their distance field. Formally, given a distance field $f(x), x \in \mathcal{R}^3$, the surface is the set $X = \{x \mid f(x) = 0\}$. However, this implicit representation suffers from two challenging aspects when applied to cloth untangling. First, untangling requires inside-outside information to resolve queries. Second, garments are open surfaces with holes that allow inner layers to pass through, introducing great complexity to the process of collision detection. To the best of our knowledge, we develop a first neural model of garments that addresses these challenges with an implicit representation.

To this end, we represent the garment using two fields: a *signed distance field* $f(x)$ that represents the garment surface and provides a notion of inside-outside and a *covariant field* $h(x)$ that models the volume near the openings that other garments can pass through without producing tangled configurations. We construct the signed distance field by calculating the euclidean distance to the surface and computing the sign as $sign((x - p) \cdot n)$ where $p$ is the closest surface point and $n$ is the normal vector at point $p$. The covariant field [8] is computed via a Hermite Radial Basis Function (HRBF) [74] fit by constraining the normals of the seams of cutout regions. Using these fields, we can detect if a point $x$ is in a tangled configuration if $f(x) < 0$ and $h(x) < 0$. Hence, points with $h(x) > 0$ are considered to be unproblematic, as they lie in the volume extending the surface holes.

A surface with holes, *e.g.*, a garment surface $g$, can be represented implicitly using this pair of fields, $g(x) = (f(x), h(x))$. We use a neural model with parameters $\theta_{\text{fields}}$ to represent these two fields. Moreover, using a neural model allows us to further parameterize the surface based on an additional code $\beta$, which yields a model $g(x, \beta, \theta_{\text{fields}})$. In our case, we parameterize garment surfaces as a function of body shape [36], but the implementation could be extended to include other codes such as body pose [57, 50, 23] or garment design parameters [68, 60], as in previous neural VTO models.

We train the neural surface model in a supervised way, with loss terms for errors in the fields and their gradients with respect to ground-truth data. Additionally, we encode points $x$ using Fourier Features [67], but omit it in the text to simplify the notation. Formally:

$$\theta_{\text{fields}} = \arg\min \quad \mathcal{L}_f + \mathcal{L}_h + \lambda \left( \mathcal{L}_{\frac{\partial f}{\partial x}} + \mathcal{L}_{\frac{\partial h}{\partial x}} \right). \tag{1}$$

$$\mathcal{L}_f = \sum_\beta \sum_x |f(x, \beta, \theta_{\text{fields}}) - f_{\text{GT}}(x, \beta)| \tag{2}$$

$$\mathcal{L}_h = \sum_\beta \sum_x |h(x, \beta, \theta_{\text{fields}}) - h_{\text{GT}}(x, \beta)| \tag{3}$$

$$\mathcal{L}_{\frac{\partial f}{\partial x}} = \sum_\beta \sum_x \left\| \tfrac{\partial f}{\partial x}(x, \beta, \theta_{\text{fields}}) - \tfrac{\partial f_{\text{GT}}}{\partial x}(x, \beta) \right\|_1 \tag{4}$$

$$\mathcal{L}_{\frac{\partial h}{\partial x}} = \sum_\beta \sum_x \left\| \tfrac{\partial h}{\partial x}(x, \beta, \theta_{\text{fields}}) - \tfrac{\partial h_{\text{GT}}}{\partial x}(x, \beta) \right\|_1 \tag{5}$$

In the supplementary document we provide additional details about the architecture of the neural network, training hyperparameters, and our strategy to sample $\beta$ and $x$.

### 3.2 Neural Untangling

Let us take as input $N$ possibly colliding implicit surfaces $\{X_i^*\}$ defined by pairs of signed-distance and covariance fields $f_i^*(x), h_i^*(x)$, respectively. Note that the surfaces can be further parameterized by a code $\beta$ as discussed above. However, we drop this parameterization in this section, as it does not affect the untangling operation. The subindex $i$ denotes the order in which the surfaces should be layered, with surface $i + 1$ above, *i.e.*, outside, surface $i$.

We perform untangling by outputting $N$ implicit surfaces $\{X_i\}$ defined by signed distance fields $f_i(x)$. We seek surfaces that are as close as possible to the input surfaces, but remain collision free.

Thanks to the implicit surface representation, untangling can be formulated as a local operation on the field values at positions $x \in \mathcal{R}^3$. Formally, untangling takes as input two vectors of field values $\mathbf{f}^* = (f_1^*, f_2^*, \ldots f_N^*) \in \mathcal{R}^N, \mathbf{h}^* = (h_1^*, h_2^*, \ldots h_N^*) \in \mathcal{R}^N$, with components $f_i^* = f_i^*(x), h_i^* = h_i^*(x)$, and it outputs a vector of field values $\mathbf{f} = (f_1, f_2, \ldots f_N) \in \mathcal{R}^N$. We denote the local untangling operation as $\mathbf{f} = \mathcal{O}(\mathbf{f}^*, \mathbf{h}^*)$. Applying the local untangling operation at positions $x$, we obtain the untangled layered field representation $\mathbf{f}(x)$, as shown schematically in Fig. 1.

The definition of the local untangling operation borrows from the method by Buffet *et al.* [8]. We define this operation as the following optimization:

$$\mathbf{f} = \mathcal{O}(\mathbf{f}^*, \mathbf{h}^*) = \arg\min \|\mathbf{f} - \mathbf{f}^*\|_2^2 + \sum_i \sum_{j>i} H(f_i, h_i^*, f_j, h_j^*), \tag{6}$$

$$H(f_i, h_i^*, f_j, h_j^*) = \begin{cases} \infty & \text{if } f_i < 0 \text{ and } h_i^* < 0 \text{ and } f_j > 0 \text{ and } h_j^* < 0 \\ 0 & \text{otherwise} \end{cases}$$

In a nutshell, this optimization returns the closest collision-free field values. The collision loss $H()$ penalizes the total loss when a point is outside the top surface $j$ and inside the bottom surface $i$. Buffet *et al.* [8] designed an algorithm of complexity $O(N^3)$ to solve the local untangling operation.

Instead, we propose a neural model with parameters $\theta_{\text{untangl}}$ that *learns* the untangling operation. Then, $\mathcal{O}(\theta_{\text{untangl}})$ can be regarded as a projection operator that projects colliding field values to the closest collision-free values, the ULNeFs $\mathbf{f}$. Importantly, note that this neural projection operator is trained only once for any arbitrary combination of $N$ surfaces, as it operates on the field values, not on the actual surfaces. Hence, once trained, this model naturally generalizes to unseen garments at train time.

We train the neural projection operator by randomly sampling values of $\mathbf{f}^* \in \mathcal{R}^N$ from $\mathcal{U}(-0.2, 1.5)$ and $\mathbf{h}^* \in \mathcal{R}^N$ from $\mathcal{U}(-1.0, 1.0)$. For each pair of $\mathbf{f}^*$ and $\mathbf{h}^*$ we compute ground-truth values of the untangled surfaces $\mathbf{f}_{\text{GT}}$ using the method by Buffet *et al.* [8].

$$\theta_{\text{untangl}} = \arg\min \quad \sum \|\mathcal{O}(\mathbf{f}^*, \mathbf{h}^*, \theta_{\text{untangl}}) - \mathbf{f}_{\text{GT}}\|_1 \tag{7}$$

Please check the supplementary document for details about training data, architecture and parameters.

## 4 Mix-and-Match VTO

Using the ULNeFs presented in the previous section, we now describe how we solve the problem of mix-and-match VTO. The input to the VTO problem consists of neural parametric models of garments

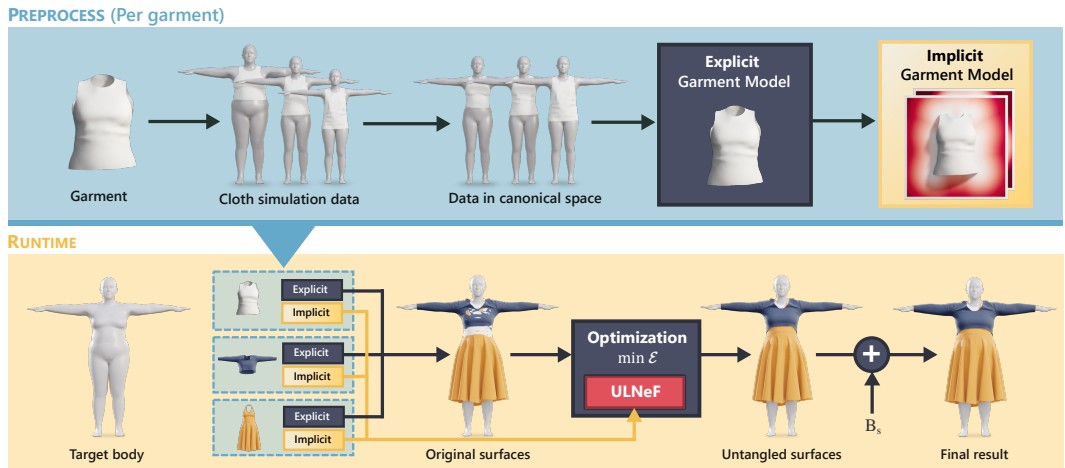

Figure 2: Pipeline of our method for mix-and-match VTO. We first preprocess a dataset of garments by simulating each of them in a variety of human shapes. Then, we transform garments into a canonical space, and learn shape-dependent explicit and implicit models. At runtime, we infer explicit and implicit shape-dependent garment deformations, use ULNeF to untangle the implicit representations, and optimize the explicit surfaces to fit into the resulting untangled fields.

trained for each garment in isolation, together with a value of the parametric code to be evaluated (in our case, body shape β). We describe an optimization problem that takes the per-garment models and finds untangled collision-free garments with minimal deformation. The central ingredient of this optimization is the fast evaluation of ULNeFs, as we search for the optimum. Figure 2 depicts our mix-and-match pipeline.

To ease the problem, similar to other VTO methods [59, 50, 5], all garment deformations are represented in a canonical body space, free of shape or pose deformations. We start by describing the body model and the canonical-space garment deformation, and then present the optimization of untangled garments.

## 4.1 Explicit Garment Model

Our explicit garment model builds on top of the parametric SMPL body model [36] and uses an explicit geometry (*i.e.,* vertices and triangles) to represent garment deformations. SMPL defines a template surface $\mathbf{T}_B$, local shape- and pose-dependent deformations with respect to this template, and provides a skinning transformation to world space. We have limited our implementation to shape-dependent transformations; therefore, we omit pose and skinning transformations in the following. In SMPL, a point on the body surface $M_B$ for shape β is defined as:

$$M_B(x, β) = x + B_s(x, β), \quad x \in \mathbf{T}_B, \tag{8}$$

where $B_s$ represents a deformation modeled using shape-dependent blend shapes.

To represent neural garment deformations, we follow a similar formulation and, specifically, we use the canonical garment model introduced by Santesteban *et al.* [59]. This canonical model diffuses surface body properties (in our case, the shape-dependent blend-shape deformations $\widetilde{B}_s$) beyond the body surface, to any point in $\mathcal{R}^3$. This diffusion strategy allows us to retrieve accurate per-point shape-dependent deformations. Hence, following the same formulation as for the body surface in (8), a point on the garment surface $M_G$ is obtained by transforming the garment in canonical space $\mathbf{X}$:

$$M_G(x, β) = x + \widetilde{B}_s(x, β), \quad x \in \mathbf{X}(β), \tag{9}$$

where $\widetilde{B}_s(x, β)$ is the diffused shape blendshape that outputs per-point 3D deformations as a function of the point $x \in \mathcal{R}^3$ and shape parameter β. Notice that $\mathbf{X}(β)$ is the deformed garment (*i.e.*, it encodes shape-dependent wrinkles), but it does not include the body shape deformations because it is encoded in our canonical space. Hence, $\widetilde{B}_s(x, β)$ is needed to add body shape deformations. This is visualized in Figure 2, bottom row, after the optimization step the untangled surfaces are shown in

the canonical space, and then $\widetilde{B}_s(x, \beta)$ is added to bring the deformations into the full space. See supplementary document for more details.

The garment model is capable of producing accurate and fast deformations of a single garment, but combining the output of multiple models results in deeply tangled surfaces. In the following section, we describe our approach to solve this issue by leveraging ULNeFs.

## 4.2 Optimization of Untangled Garments

To obtain untangled garment surfaces $\{\mathbf{X}_i(\beta)\}$ for a specific shape code $\beta$, we reconstruct the zero-sets of ULNeFs. Note that ULNeFs define implicit surfaces $\{X_i(\beta)\}$, and here we search for explicit mesh-based discretizations. Moreover, since ULNeFs are defined by per-garment neural parametric fields, we use the method presented in Section 3.1 to train an implicit equivalent of each explicit garment model.

To reconstruct the zero-sets of ULNeFs, first we initialize possibly colliding garments $\{\mathbf{X}_i^*(\beta)\}$ using the per-garment explicit models. We have observed that just projecting mesh vertices to the zero-sets could yield large triangle distortions. Therefore, when searching for the untangled garment surfaces, we add a penalty term to minimize triangle distortion. Formally, we obtain each untangled garment surface by solving the following optimization:

$$\mathbf{X}_i(\beta) = \arg\min \quad \mathcal{E}_{\text{projection}} + \omega\, \mathcal{E}_{\text{strain}} \tag{10}$$

$$\mathcal{E}_{\text{projection}} = \sum_{x \in \mathbf{X}_i(\beta)} f_i(x, \beta)^2, \tag{11}$$

$$\mathcal{E}_{\text{strain}} = \sum_{T \in \mathbf{X}_i(\beta)} \left\| \tfrac{1}{2}(F(T)^\top F(T) - \mathbf{I}) \right\|_2^2. \tag{12}$$

In the $\mathcal{E}_{\text{projection}}$ term, we evaluate the untangled field $f_i$ for all vertices $x$ in the garment mesh. Note that this requires first evaluating per-garment fields, followed by the neural projection, as shown in Figure 1. In the $\mathcal{E}_{\text{strain}}$ term, we evaluate the squared Frobenius norm of Green strain for all triangles $T$ in the garment mesh, with $F$ the deformation gradient.

We solve the optimization (10) using L-BFGS. We have observed that initialization with the per-garment meshes $\{\mathbf{X}_i^*(\beta)\}$ is key for fast convergence of the optimization. Note also that the gradient computation requires the gradient of the ULNeFs, which is easily obtained thanks to the automatic differentiation capabilities of machine learning frameworks.

# 5 Evaluation

## 5.1 Quantitative Evaluation

In Table 1 we present an ablation study of the different terms and encodings used to train the implicit representation for open surfaces described in Section 3.1. For each ablation, we show the error of the two fields used in our representation. Results demonstrate that both the encoding of input points with Fourier Features [67] and the supervision of the gradients contribute to training the model.

Table 2 evaluates the runtime performance of our approach. Specifically, we compare the evaluation time of the untangling operator of Buffet *et al.,* [8] (*i.e.,* solving Equation 6) vs. a forward pass of our learned projection operator. It demonstrates that for complex outfits with thousands of vertices (the outfits shown in Figure 3 range from 15k to 30k vertices), our approach runs up to two order of magnitude faster. Similarly, our formulation to evaluate the covariant fields $f$ and $h$ is also significantly faster.

## 5.2 Qualitative Evaluation

In Figure 4 we present a qualitative ablation of study of the different terms used to learn our implicit garment model described in in Section 3.1. We show that using Fourier Features [67] to encode points, as well as supervising the gradient loss is required to obtain a accurate neural fields to encode detailed garments.

|  | Ours | | W/o Fourier feats. [67] | | W/o gradient supervision | |
|---|---|---|---|---|---|---|
|  | $f$ | $h$ | $f$ | $h$ | $f$ | $h$ |
| Error (T-shirt) | **1.1mm** | **0.8mm** | 2.0mm | 1.0mm | 5.3mm | 0.9mm |
| Error (Dress) | **1.3mm** | 2.0mm | 1.6mm | 2.0mm | 6.6mm | **1.9mm** |

Table 1: Ablation study of the different terms of our implicit surface model described in Section 3.1.

|  | Untangling operator | | Field evaluation | |
|---|---|---|---|---|
| Nº vertices | Buffet *et al.* [8] | Ours | Buffet *et al.* [8] | Ours |
| 1 | 0.04 ms | 0.24 ms | 0.08 ms | 0.36 ms |
| 5000 | 81.6 ms | 0.68 ms | 2.08 ms | 0.77 ms |
| 15000 | 238.5 ms | 1.74 ms | 6.02 ms | 2.00 ms |
| 30000 | 508.0 ms | 3.35 ms | 12.1 ms | 3.92 ms |

Table 2: Comparison of runtime performance of the main components of ULNeF. We use the authors' implementation to compare the performance of the untangling operator, and an efficient GPU reimplementation to compare the fields. This comparison was conducted in a regular desktop PC equipped with an AMD Ryzen 7 2700 CPU, an Nvidia GTX 1080 Ti GPU, and 32GB of RAM.

In Figure 3, and in the supplementary video, we show qualitative results of mix-and-match VTO. For each example, we show the entangled result that state-of-the-art VTO methods [59, 59, 50] produce when predicting the deformations for multiple garments, without any postprocess.

## 6   Conclusions

Motivated by the shortcoming of state-of-the-art data-driven methods for virtual try-on (VTO) (*i.e.*, the inability to deal with multiple garments), we have identified fundamental limitations in emerging surface representations based on neural fields [75]. These limitations effectively prevent current neural field methods from modelling common scenarios such as contact and interaction between objects.

To address this, we have presented Untangled Layered Neural Fields (ULNeF), a novel neural approach to project entangled implicit surfaces to an untangled configuration. The zero-set of the projected fields is collision-free and minimizes the difference with respect to the input surfaces (*i.e.*, maintain the fine-scale details). Importantly, ULNeFs generalize to neural fields unseen at train time. Additionally, they are highly efficient to evaluate since they only require a forward pass (*i.e.,* a projection, and no iterative optimization) to find the untangled fields.

We have demonstrated the applicability of ULNeFs in mix-and-match VTO, where we untangle the implicit representations of parametric garments, and optimize the garment explicit surfaces driven by the untangled fields. This enables for first time interactive, accurate, and collision-free combinations of 3D garments, while being able to manipulate the underlying body shape.

**Discussion.**   In Section 4 we showcased a downstream task with ULNeFs consisting on mix-and-match VTO. We assume that a requirement for such task is to untangle the input garments *while keeping the original mesh topology*, and show that this can be achieved by fitting the input meshes to the untangled fields (Section 4.2). However, we want to stress that if a fixed mesh topology is not a requirement, marching cubes could be used directly after projecting the neural fields with ULNeF.

All in all, we believe ULNeF makes an important step towards modeling interactions of neural fields. Future research could explore the use of ULNeF in other scenarios that need to account for contact (*e.g.,* hand-object interaction) and are currently limited by explicit surface models.

**Limitations.**   The proposed approach for VTO using ULNeFs has only been validated with garments in T-pose. The root of this limitation is the difficulty in extending the for-

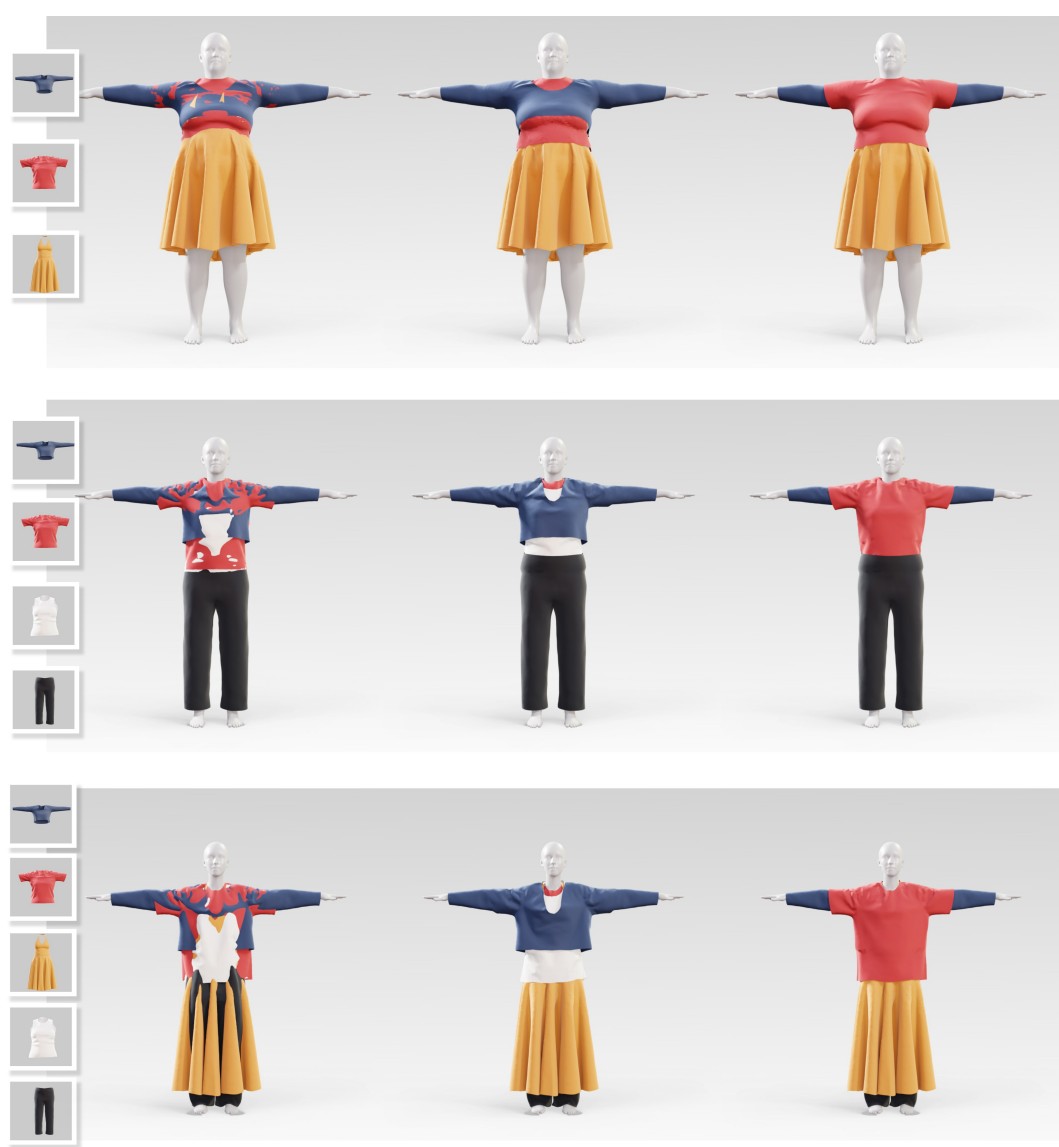

Figure 3: Given a set of garments (left insets), existing VTO methods [57] infer their fit into a target body shape but produce a heavily entangled results (left). In contrast, ULNeF untangles the garments by directly projecting their neural fields into an collision-free configuration. Since ULNeF allows to specify the desired order, different outfits can be created (center and right).

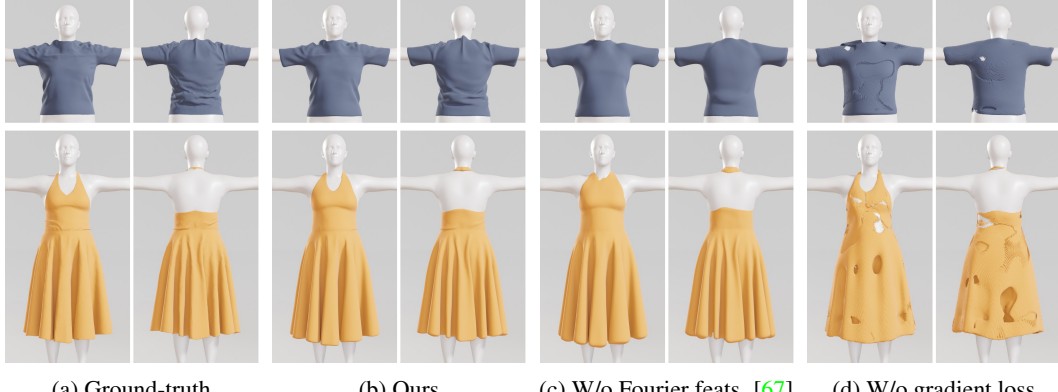

| (a) Ground-truth | (b) Ours | (c) W/o Fourier feats. [67] | (d) W/o gradient loss |

Figure 4: Qualitative ablation study of our implicit garment model described in Section 3.1. For this particular figure, we use Marching Cubes to extract the surface.

mulation based on covariant fields to more complex poses, but we believe that representing garments in an unposed canonical space could be helpful to circumvent this issue.

Similarly, garments with highly curved boundaries (e.g., neck or sleeves) can be problematic because the computed covariant field might encode wrong surface semantics. Notice that our formulation based on covariant fields is used to approximate the signed distance values around the open areas (*i.e.*, semantic information about the position of a given point with respect to the surface). Therefore, since the estimation of such covariant fields rely on the normals of the vertices, areas with high curvature (*i.e.*, non-smooth normals) can lead to wrong fields. A potential solution to this issue could be investigating an alternative to our covariant fields that leads to a smoother representation.

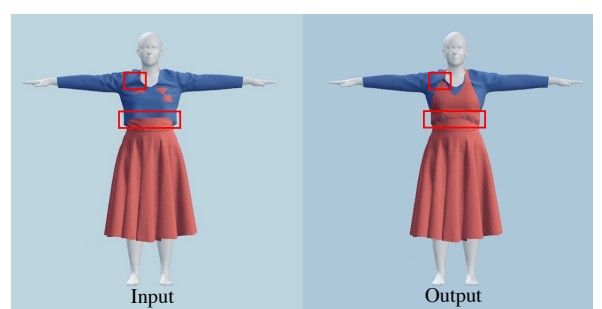

Figure 5: Limitations. ULNeF struggles with input configurations (left) with vertices located too far from the untangling area or in the border of the garment. Untangled results (right) can exhibit residual collisions in such specific areas.

Additionally, although we achieve a significant speed up compared to previous works [8], our overall runtime is in the order of 200ms per frame. While this is good enough for interactive mix-and-match VTO applications in static pose, it falls short of producing real-time animations of untangled outfits. Hence, further improvements towards reducing the computational cost remain open avenues for future works.

**Acknowledgments.** The work was funded in part by the European Research Council (ERC Consolidator Grant no. 772738 TouchDesign) and Spanish Ministry of Science (RTI2018-098694-B-I00 VizLearning).

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
