# OpenReview forum: "ULNeF: Untangled Layered Neural Fields for Mix-and-Match Virtual Try-On"
_NeurIPS.cc/2022/Conference — NeurIPS 2022 Accept_

### Official Review · Reviewer_q8FC · 2022-07-07

**Rating:** 7
**Confidence:** 4
**Soundness:** 4 excellent
**Presentation:** 4 excellent
**Contribution:** 4 excellent

**Summary:**

This work presents a novel approach to mix-and-match 3D virtual try-on.  The proposed model is able to untangle layered neural fields (how garments are represented) by introducing a projection operator that works directly on neural implicit representations. The composition of the layered neural field with the neural projection is what the authors refer to as a ULNeF.  At runtime, the model is able to take as input a representation of the body shape and optimize for a layering of the garments by using ULNeFs to resolve collisions.

**Questions:**

Are there any failure cases with the ULNeF model? For example, are there any types of clothing with certain features that would prove problematic for this approach?

What would happen if the ordering of the garments did not make sense? For example, if one were to layer the pants on top of the dress?

[159-161] Have the authors performed any experiments with parameterizing additional codes, other than body shape?


**Limitations:**

The authors have acknowledged that the proposed method has only been demonstrated on a single pose (T-pose).  The work is based on simulated models, and does not involve real humans or garments, so potential negative societal impact is lessened..

**Strengths And Weaknesses:**

Strengths:
* Mix-and-match is a very challenging problem in VTO, and the authors demonstrate results on up to 7 layers of garments, which is quite impressive.

* ULNeFs could be applicable to other object-object interaction problems, as the authors point out.

* The neural model introduced that learns the untangling operation results in significant efficiency gains (“up to two orders of magnitude faster” [258-259]).

* The authors provide sufficient details about implementation, such that this work would be reproducible by others.

Weaknesses:
* No comprehensive quantitative evaluation is performed to measure the quality of the mix-and-match results, besides the ablation study of terms and encodings used for training the implicit surface model.  Only a handful of qualitative examples are shown, though the video in the supplemental material shows a compelling interactive demo.

* The authors acknowledge that they have only validated their work on a single pose, using simulated models and clothing.

---

> ### Author Response · Authors · 2022-08-01
> **Answers to Reviewer q8FC**
>
> Thank you very much for you review. Here we clarify the few points that were not clear, and answer your questions.
>
> > **Are there any failure cases with the ULNeF model? For example, are there any types of clothing with certain features that would prove problematic for this approach?**
>
> Garments with highly curved boundaries (e.g., neck or sleeves) can be problematic because the computed covariant field might encode wrong surface semantics. Notice that our formulation based on covariant fields is used to approximate the signed distance values around the open areas (i.e., semantic information about the position of a given point with respect to the surface). Therefore, since the estimation of such covariant fields rely on the normals of the vertices (see lines 152-155), areas with high curvature (i.e., non-smooth normals) can lead to wrong fields. A potential solution to this issue could be investigating an alternative to our covariant fields that leads to a smoother representation.
>
> > **What would happen if the ordering of the garments did not make sense? For example, if one were to layer the pants on top of the dress?**
>
> We have not tried such extreme cases, which could be discarded by checking geometric heuristics based on genus or similar properties of the garments. Nonetheless, we think our current optimization defined in Eq 10 could fall into a local minimum, and it would be very hard to reach the global minimum, with the dress squeezed all the way to the waist to avoid intersections with the pants.
>
> > **Have the authors performed any experiments with parameterizing additional codes, other than body shape?**
>
> As we discuss in our limitation section [l294], we have validated ULNeF for T-pose only (i.e., using only shape parameters). The key challenge in adding additional parameters (e.g., pose) lies in computing covariant fields for deformed posed garments, since the resulting HRBF might not encode the required surface semantic information (i.e. approximate distance to the surface in open areas). Nonetheless, we envision that such an issue could be circumvented by untangling posed garments in a canonical pose, and re-pose the resulting surfaces similar to the way Santesteban et al. [CVPR 21] solve body-garment collisions.

---

### Official Review · Reviewer_U9t8 · 2022-07-12

**Rating:** 7
**Confidence:** 4
**Ethics Flag:** Yes
**Soundness:** 4 excellent
**Presentation:** 3 good
**Contribution:** 4 excellent

**Summary:**

This paper presents ULNef, which leverages layered neural fields to create collision-free clothing surfaces for mix-and-match virtual try-on.

**Questions:**

- I encourage the authors to add visual results of ablations and failures cases.

- Comparing with pure cloth simulation methods would be better provided.

**Limitations:**

The authors have discussed several limitations of the proposed method. I am interested in looking at some visual failure cases and more in-depth analysis.

**Strengths And Weaknesses:**

*Strength*

- This paper aims to resolve an interesting problem that how to fit multi-layer garments to a target body shape while avoiding colliding among garment layers. The authors propose a novel model that untangles the layered neural fields with neural implicit representations (i.e., a signed distance field and a covariant field). This potentially introduces a new paradigm for solving related problems.

- The visual results clearly verify the effectiveness of the proposed method.

- The way of obtaining SDF from open surfaces is interesting and has proven to be effective.

- I like the pipeline of solving layered VTO and the results are promising.

*Weaknesses*

- Only a quantitative ablation study is given in Table 1. A visual comparison should be given for understanding how the Fourier features and gradient supervision contribute to reducing the garment collision qualitatively.

- The authors briefly discussed the limitations without showing visual failure examples. In my opinion, illustrating some failure cases (for example, how does the method work on a posed body, or extreme body shapes say very large or small $\beta$? Will the results look bad?) can help to guide future research directions.

- Can the mix-and-match VTO be directly achieved by using cloth simulation? Is it possible to compare the proposed method with its simulation counterparts?

- Minor: Figure 1 could be more informative. For now, it is hard to make sense out of this simple diagram.

---

> ### Author Response · Authors · 2022-08-01
> **Answers to Reviewer U9t8**
>
> Thank you very much for you review. Here we clarify the few points that were not clear, and answer your questions.
>
> > **I encourage the authors to add visual results of ablations and failures cases.**
>
> We agree that adding qualitative visualization of the ablations and failure cases would be very informative. Due to time constraints for personal reasons we cannot provide them during the rebuttal phase, but we will include failure cases and qualitative ablations in our final version.
>
> > **Comparing with pure cloth simulation methods would be better provided.**
>
> We kindly remark to the reviewer that cloth simulation algorithms require an initial configuration that is collision-free, and then they enforce collision-free deformations thanks to continuous collision detection and constrained contact mechanics. Therefore, these algorithms cannot resolve any of the initial tangled configurations shown in Fig. 3 (left). Nonetheless, the output of ULNeF could be used as initial configuration in a cloth simulator to animate the avatar.
>
> > **Figure 1 could be more informative. For now, it is hard to make sense out of this simple diagram.**
>
> We agree Figure 1 could be more informative, and will improve it by adding visualizations and more descriptive labels.
>
> > **Results for extreme body shapes, say very large or small $\beta$?**
>
> Since ULNeF is a local operator, *global* extreme shapes produced by a very large $\beta$ would not directly affect our performance. However, if the associated explicit and/or implicit representation of the garment (i.e., the preprocessed representation, as shown in top Fig.2) is not capable of representing such extreme case correctly, ULNeF would naturally suffer from the poor encoding. We are happy to add a discussion about this in the final version.

---

### Official Review · Reviewer_LMYH · 2022-07-15

**Rating:** 6
**Confidence:** 4
**Soundness:** 3 good
**Presentation:** 3 good
**Contribution:** 3 good

**Summary:**

Based on recent success on neural cloth simulation for a single garment, the authors extend previous works to solve multi-layer cloth prediction while avoiding collisions. The author leverages the untangling power of the implicit surface model from Buffet et al. and neural cloth representation from Santesteban et al., designing a new neural untangling layer directly predicting collision-resolved multi-layer cloth pieces. The authors have shown a cool demo in the video that they can directly predict multi-layer cloth on the human bodies while tuning the human height and width.

**Questions:**

1. Sec 4.2 and Sec. 3.2 both are inspired by numerical optimization, why not also trained a network to replace the optimization in  Sec. 4.2?

2. In Eq. 9 the X(beta) is already dependent on shape parameters, why also need a diffusion step? Directly train a network to replace Mg(x, beta) maybe better?

**Limitations:**

The work is only evaluated on simplified canonical space.

**Strengths And Weaknesses:**

Pro:

1. To extend a neural network-based virtual try-on system to a multi-layer cloth system is very novel and ambitious. The method utilizes a neural implicit representation of cloth and uses an optimization-inspired method to untangle different cloth pieces. The authors have shown cool visualization results that they can handle up to ~5 layers of cloth pieces stacking on a human body. In the ablation study, they also include results that using Fourier features and gradient loss can generate a better implicit representation of the garment model.

2. Another exciting part of this paper is the authors have built a comprehensive pipeline for the mix-and-match VTO task. From neural explicit/implicit representation to untangling, to find the final zero-set surface using optimization. It's not easy to fuse several challenging components to a final runnable pipeline.

Con:

1. Sec 4.2 about optimization of untangled garments is not well-written and very unclear. In line 242, they mentioned that "evaluating per-garment fields, followed by the neural projection (untangling I guess?)", not sure if it is only queried once to find the zero-level surface or done repeatedly during the optimization. Also, it is unclear if it is solving the {X_i} with all the pieces together or only finding layer by layer. I suggest the authors add an algorithm section to detailly include how they combine L-BFGS and the untangling operator.

2. Current results are limited by the evaluation, the authors simplified the problem to T-pose (with only height, and width), without pose parameters.

Some related work:
Collision avoidance based on neural network and optimization:
 LCollision: Fast Generation of Collision-Free Human Poses using Learned Non-Penetration Constraints
Q Tan, Z Pan, D Manocha
AAAI Conference on Artificial Intelligence

---

> ### Author Response · Authors · 2022-08-01
> **Answers to Reviewer LMYH**
>
> Thank you very much for you review. Here we clarify the few points that were not clear, and answer your questions.
>
> > **Sec 4.2 about optimization of untangled garments is not well-written and very unclear. In line 242, they mentioned that "evaluating per-garment fields, followed by the neural projection (untangling I guess?)", not sure if it is only queried once to find the zero-level surface or done repeatedly during the optimization. Also, it is unclear if it is solving the {X_i} with all the pieces together or only finding layer by layer. I suggest the authors add an algorithm section to detailly include how they combine L-BFGS and the untangling operator.**
>
> As indicated by Eq. 10 in Sec. 4.2, each garment can be optimized independently. The ULNeFs ensure that each garment converges independently to a configuration where the field values are collision-free. Also, note that eq. 11 is part of the objective function; therefore, the ULNeFs must be evaluated on each iteration of the optimization and for every cloth vertex, until convergence. L-BFGS simply requires evaluating on each iteration the gradient of the objective function. Concerning the ULNeFs part of eq. 11, this is easy thanks to the differentiability of the neural model. We will provide an algorithmic exposition of the optimization for clarity.
>
> > **Some related work: Collision avoidance based on neural network and optimization: LCollision: Fast Generation of Collision-Free Human Poses using Learned Non-Penetration Constraints Q Tan, Z Pan, D Manocha AAAI Conference on Artificial Intelligence**
>
> Thanks for the suggestion, we agree this work is relevant and will add a discussion about it into our final version.
>
> > **Sec 4.2 and Sec. 3.2 both are inspired by numerical optimization, why not also train a network to replace the optimization in Sec. 4.2?**
>
> Please note that the optimization formulated in Eq.11 (Sec 4.2) is outfit-specific: the projection term requires $\mathbf{f}$, which is a number-of-layer specific untangled field; and the strain term is global since it uses all triangles of a garment; hence, it would be nearly impossible to train a network that generalizes to new combinations due to the intractable combinatorial problem. In contrast, the untangling operator – the key contribution of our work – defined in Sec 3.2 (Eq. 7) is a local operator that naturally generalizes to unseen garments.
>
> > **In Eq. 9 the X(beta) is already dependent on shape parameters, why also need a diffusion step? Directly train a network to replace Mg(x, beta) maybe better?**
>
> Thank you for asking this, we’ve realized that the wording used to define X(\beta) after Eq. 9 is not accurate. To clarify: X(\beta) *is* shape-dependent, because it encodes shape-dependent wrinkles, but it does not include the body shape deformations because it is encoded in our canonical space. Hence, B_s is needed to add body shape deformations. This is visualized in Fig 2, bottom row, after the optimization step the untangled surfaces are shown in the canonical space, and then B_s is added to bring the deformations into the full space. We will rephrase lines 222-225 to clarify this in the final version.

---

### Official Review · Reviewer_Czdg · 2022-07-15

**Rating:** 5
**Confidence:** 3
**Soundness:** 2 fair
**Presentation:** 3 good
**Contribution:** 2 fair

**Summary:**

The paper proposed a framework for  virtual try-on with multiple garments. The idea is first encode the the open surface implicitly with signed distance field and covariant field, and to combine multiple garment with collision free condition, the paper further proposed a formulation to use neural network to predict the untangle layered implicit surfaces, comparing with previous work (Buffet et al. [8]) which uses optimization process to achieve collision free. The final algorithm is formalizing the VTO as an optimization problem (Eq. 10), which needs to achieve both collision free and minimize distortions of the triangles. Qualitatively, the results achieves better performance comparing with previous works on VTO.

**Questions:**

See Weakness section above

**Ethics Review Area:**

["I don’t know"]

**Limitations:**

See Weakness section above

**Strengths And Weaknesses:**

Strength:

1. The paper is well written and easy to follow.

2. The final performance on VTO looks better than baselines, qualitatively speaking.


Weakness

1. Novelty. As summarized above (and also in the paper), there're two new things proposed in the paper: a) A neural field formulation to encode open surfaces, b)  utilizing a network to predict the untangled distance field, instead of optimization procedure. However, are these two novel? Why using them is correct choice? Regarding the encoding of open surfaces, there're already many published works [a, b], why the proposed method is better than these work? The paper didn't provide an explanation on this, nor comparisons. Regarding using a network to predict untangled distance field, this is a new approach to me, however, Buffet et al. [8] can solve the fundamental limitation mentioned in the abstract "the interaction between layered neural fields.", to me, the benefit is only using a neural network is much faster than [8], are there other advantages?

2. Limitation. There are a bunch of limitation of proposed methods:

- The experiment is only designed for canonical pose, extending to arbitrary poses is not clear (It's not as simple as mentioned in the Line 160-161), different poses will make the garment fitting more complicated,  and ULNef module might also have generalization issue.
- The ULNef is limited to a fixed number of garment, the input is N dimension vector and output is also N dimension vector, how does it generalize to different number of garments?
- The model would require to fit each module for each garment, how could this scale up to real world application, where we can have thousands of different clothes.
- Generalization of ULNef. It's hard to say whether ULNef can generalize to different garments, e.g. trained one one set of garment, but tested on a different set of garments (there's also no analysis of this)

3. Expeirment. The experiment has many problems:

- The biggest issue for experiment is the model is only trained and tested on 5 garments (See Supplement Sec 1), this is very limited in evaluating the performance to check the whether the result can generalize in a larger scale.
- It would be better to ablate each module proposed in the framework. I think there are missing at least two baselines for comparisons: i) To show the effect of proposed idea of encoding open surfaces, it would be better to compare at least one of [a], [b]. ii) To show the effect of ULNef, one baseline is not using a neural network to predict $\textbf{f}$, but use [8] to get $\textbf{f}$


In summary, this paper has many problems in both the method and experiment side, and I think it's not ready for publication at current status, so I vote for a reject.

[a] NEURAL UNSIGNED DISTANCE FIELDS FOR IMPLICIT FUNCTION LEARNING
Julian Chibane, Aymen Mir, Gerard Pons-Moll
NeurIPS 2020

[b] 3PSDF: Three-Pole Signed Distance Function for Learning Surfaces with Arbitrary Topologies
Weikai Chen, Cheng Lin, Weiyang Li, Bo Yang
CVPR 2022

[POST REBUTTAL]
After reading the rebuttal and other reviews, I'm convinced of the results provided in the paper, but I still do not think the novelty is great enough for neurips, so I only raised my score to boardline accept.

---

> ### Author Response · Authors · 2022-08-01
> **Answers to Reviewer Czdg**
>
> Thank you very much for you review. Here we address your questions to clarify the relevance of our work.
>
> > **Regarding the encoding of open surfaces, there are already many published works [[a](https://proceedings.neurips.cc/paper/2020/file/f69e505b08403ad2298b9f262659929a-Paper.pdf), [b](https://openaccess.thecvf.com/content/CVPR2022/papers/Chen_3PSDF_Three-Pole_Signed_Distance_Function_for_Learning_Surfaces_With_Arbitrary_CVPR_2022_paper.pdf)], why the proposed method is better than these work?**
>
> We would like to kindly remark that our covariant field formulation creates meaningful semantic information on surfaces about how open areas have to be handled. That is, it approximates the signed distance values around the open areas, and as such gives information that  alternative formulations do not directly provide. For example, your suggested work [[a]](https://proceedings.neurips.cc/paper/2020/file/f69e505b08403ad2298b9f262659929a-Paper.pdf) computes unsigned distances, which complicates the extraction of isosurface, and does not allow for inside/outside queries to model collisions (as we discuss in line 108). [[b]](https://openaccess.thecvf.com/content/CVPR2022/papers/Chen_3PSDF_Three-Pole_Signed_Distance_Function_for_Learning_Surfaces_With_Arbitrary_CVPR_2022_paper.pdf) gives signs to points around open surfaces, but it also uses a novel null sign in some areas, which does not fulfill the sign requirements of our untangling as defined in Eq 6. Furthermore, we’d like to point out that [[b]](https://openaccess.thecvf.com/content/CVPR2022/papers/Chen_3PSDF_Three-Pole_Signed_Distance_Function_for_Learning_Surfaces_With_Arbitrary_CVPR_2022_paper.pdf) is a CVPR 22 paper that was officially published after the NeurIPS deadline, hence it is considered concurrent work. We are happy to incorporate these citations and this discussion in the final version.
>
> > **Regarding using a network to predict untangled distance field, this is a new approach to me, however, Buffet et al. [8] can solve the fundamental limitation mentioned in the abstract "the interaction between layered neural fields.", to me, the benefit is only using a neural network is much faster than [8], are there other advantages?**
>
> We kindly remark that our approach runs up to 2 orders of magnitude faster than the method by Buffet et al. Such a speed up is crucial for applications such as virtual try on, where a user expects an interactive experience. On top of this speed up, ULNeF comes with additional advantages:
> 1. our learned untangling operator is trained only once for any arbitrary combination of N garments, and it naturally generalizes to new garments because it operates on the field values, not the actual surfaces. This can be seen in Eq. 6, where covariant field values h and f depend only on values at position $x$ (i.e., do not depend on the entire geometry of the garment);
> 2. our neural formulation for untangling, defined in Eq. 6, is defined entirely using implicit surfaces, while the analogous operation from Buffet et al., formulated through an iterative optimization approach, requires an explicit representation to perform the untangling. As we highlight in lines 285-289, our mesh optimization step from Eq 10 is only required if explicit mesh topology should be preserved. We believe this is an important benefit over Buffet et al. method.
>
> > **The experiment is only designed for canonical pose, extending to arbitrary poses is not clear (It's not as simple as mentioned in the Line 160-161), different poses will make the garment fitting more complicated, and ULNef module might also have generalization issue**
>
> While we acknowledge this limitation in lines 293-295, we also wish to point out that such an issue could be circumvented by untangling posed garments in a canonical pose, and re-pose the resulting surfaces. This would be similar to the way existing works like Santesteban et al. [CVPR 21] solve body-garment collisions for posed meshes.
>
> > **Generalization of ULNef. It's hard to say whether ULNef can generalize to different garments, e.g. trained one one set of garment, but tested on a different set of garments (there's also no analysis of this)**
>
> We’d like to clarify this misunderstanding: ULNeF is inherently agnostic to specific garments, and trivially generalizes to new ones. This can be seen in Eq. 6, where the covariant field values h and f depend only on values at position x (i.e., do not depend on the entire geometry of the garment, only on local values of the fields). Furthermore, the actual extraction of explicit untangled surfaces is done through an optimization step, which likewise ensures generalization.
>
> > **Limited evaluation, only 5 garments**
>
> Considering that ULNeF is a local operator (as clarified above), showcasing successful results using 5 garments of thousands of vertices over a parametric body shape is in practice evaluation our method extensively, given large number of combinations arising from the 5 garments.

---

> > ### Comment · Reviewer_Czdg · 2022-08-04
> > **Thanks for the responses**
> >
> > Thank the authors for providing a detailed rebuttal.
> >
> > I still have one more question to clarify.
> >
> >
> > In the evaluated 5 garments, it is mentioned in Appendix, Sec A. the model is trained on 5 garments. Does this mean the training and testing are running on the same garments?

---

> > > ### Author Response · Authors · 2022-08-04
> > > **Garments only used for testing**
> > >
> > > Thanks for asking this question, it is important that this becomes 100% clear. The neural untangling operator, Sec 3.2 in the paper, is completely garment agnostic. It is trained using random values, as explained in Appendix B. In this sense, all garments shown in the paper were used only for testing, none of them was used for training the neural untangling operator. Notice how, in the video, garments and their ordering are shuffled, as the neural untangling has not seen any of these garment combinations at training. Appendix A says “we train 5 garment models”, but we realize this was a bad choice of words. We should have said “we have tested the untangling using 5 garment models. For each garment, independently, we precompute the implicit surface model”. This is a per-garment operation, which does not see multi-garment outfits or any tangled configuration at any time.

---

> > > > ### Comment · Reviewer_Czdg · 2022-08-10
> > > > **Thanks for the reply!**
> > > >
> > > > I thank the author for clarification! It makes more sense to me, and I'm happy to see the performance of ULNef in this setting!
> > > >
> > > > I'm still not fully convinced by the novelty of the paper, but I'm happy to increase my score to a boardline accept.

---

### Author Response · Authors · 2022-08-01
**Thank you for the detailed and overall very positive reviews.**

We thank all reviewers for the detailed reviews and for the largely positive assessment of our work. Below, using the comment section for each review, we answer in detail all the individual questions raised.

---

### Meta-Review · Area_Chair_VK9D · 2022-08-24

**Recommendation:** Accept
**Confidence:** Certain

**Metareview:**

The reviewers found this paper novel and ambitious (neural virtual try-ons), well written, with good qualitative results, and thought the mix-n-match is a good addition.
They were concerned about certain exposition parts (section 4.2), and the experimental setup that lacked quantitative evaluation.
I encourage the authors to take into heart these comments while preparing their revision.


**Award:**

No

---

### Decision · Program_Chairs · 2022-09-14

Accept